# A Double-Edged Sword: Thioxanthenes Act on Both the Mind and the Microbiome

**DOI:** 10.3390/molecules27010196

**Published:** 2021-12-29

**Authors:** Marianne Ø. Poulsen, Sujata G. Dastidar, Debalina Sinha Roy, Shauroseni Palchoudhuri, Jette Elisabeth H. Kristiansen, Stephen J. Fey

**Affiliations:** 1Memphys Center for Biomembrane Physics, Department of Physics and Chemistry, University of Southern Denmark, 5230 Odense, Denmark; malthe@dadlnet.dk; 2Department of Microbiology, Herbicure Healthcare Bio-Herbal Research Foundation, Kolkata 700154, India; jumicrobiol@yahoo.co.in (S.G.D.); debalina.s06@gmail.com (D.S.R.); shauroseni.micro@gmail.com (S.P.); 3CelVivo ApS, 5220 Odense, Denmark; sjf@celvivo.com

**Keywords:** thioxanthenes, antipsychotic drugs, non-antibiotics

## Abstract

The rising tide of antibacterial drug resistance has given rise to the virtual elimination of numerous erstwhile antibiotics, intensifying the urgent demand for novel agents. A number of drugs have been found to possess potent antimicrobial action during the past several years and have the potential to supplement or even replace the antibiotics. Many of these ‘non-antibiotics’, as they are referred to, belong to the widely used class of neuroleptics, the phenothiazines. Another chemically and pharmacologically related class is the thioxanthenes, differing in that the aromatic N of the central phenothiazine ring has been replaced by a C atom. Such “carbon-analogues” were primarily synthesized with the hope that these would be devoid of some of the toxic effects of phenothiazines. Intensive studies on syntheses, as well as chemical and pharmacological properties of thioxanthenes, were initiated in the late 1950s. Although a rather close parallelism with respect to structure activity relationships could be observed between phenothiazines and thioxanthenes; several thioxanthenes were synthesized in pharmaceutical industries and applied for human use as neuroleptics. Antibacterial activities of thioxanthenes came to be recognized in the early 1980s in Europe. During the following years, many of these drugs were found not only to be antibacterial agents but also to possess anti-mycobacterial, antiviral (including anti-HIV and anti-SARS-CoV-2) and anti-parasitic properties. Thus, this group of drugs, which has an inhibitory effect on the growth of a wide variety of microorganisms, needs to be explored for syntheses of novel antimicrobial agents. The purpose of this review is to summarize the neuroleptic and antimicrobial properties of this exciting group of bioactive molecules with a goal of identifying potential structures worthy of future exploration.

## 1. Introduction

For the sake of clarity, we will first present the neuroleptic properties of the thioxanthenes, followed by their antimicrobial effects.

The first goal of antipsychotic medication is to minimize or eliminate the symptoms within a short period of time. Originally, antipsychotic drugs were designed and tested empirically on psychiatric patients to determine their effectiveness. The first antipsychotic drug that was primarily used as an anesthetic agent in surgery was chlorpromazine [1]. In the beginning, chlorpromazine was administered to psychiatric patients to determine its calming effects. However, it was soon realized that the drug also reduced psychosis. Antipsychotic drugs have been classified as either low or high potency based on their ability to bind to dopamine receptors and not on their effectiveness on patients. Antipsychotic drugs tend to block the action of D_2_ neuroreceptors in the dopamine pathway in the brain, resulting in reduction of the release of dopamine in the relevant synapses [2]. Thus, an antipsychotic drug should be able to also block the D_2_ receptors in the mesolimbic pathway of the brain [3,4].

Currently there are two main types of antipsychotics in use, typical and atypical. The main difference between typical and atypical antipsychotic drugs is the ability of the latter to address the negative symptoms of schizophrenia. These atypical neuroleptics, known as *newer drugs*, date back to the introduction of *clozaril* [5]. These atypical drugs usually do not cause unpleasant side effects. Atypical drugs may improve cognitive symptoms and can be effective in patients resistant to typical antipsychotic drugs. Such drugs can be active on other receptors, in addition to the dopamine receptor, and many have no extra-pyramidal side effects. Atypical antipsychotics possess an almost identical effect on D_2_ receptors but are usually more selective, targeting the intended pathway to a larger degree than other drugs.

Typical antipsychotics are sometimes referred to as *major tranquilizers*, since many of them, in large doses, can sedate and tranquilize [6].

Typical antipsychotics can be classified into three major groups (phenothiazines, butyrophenones and thioxanthenes). Of these, phenothiazines are the most widely used. Table 1 lists the chemical classification of the neuroleptics, illustrated by a selection of the most commonly prescribed drugs from each class along with their structures.

Thioxanthenes, the third group of antipsychotics, are represented in two geometric stereoisomers: *Z*- and *E*-compounds, of which the former have been shown to be more neuroleptically active [7,8]. Due to structural similarity, the antipsychotic activities of flupenthixol and clopenthixol are quite similar to the ‘piperazine group’, one of three groups in the phenothiazine class. These compounds are able to benefit psychotic patients by blocking postsynaptic dopamine receptors in the brain. Thioxanthenes also produce an alpha-adrenergic blocking effect and depress the release of a large number of hypothalamic and hypophyseal hormones [8].

Thioxanthenes were synthesized and developed with the hope of eliminating the toxic effects of chlorpromazine. The first thioxanthene that came to the market in Scandinavia in 1959 was chlorprothixene. A more potent compound, clopenthixol, was introduced in Denmark and other European countries in 1961, followed by introduction of thiothixene and flupenthixol [9].

Much like the phenothiazines, substitution in position 2 provides the intensity of neuroleptic action in thioxanthenes. It is known that the presence of a double bond in the side chain between the carbons 9 and 1 greatly increases the neuroleptic capacity of thioxanthenes. The structure may be asymmetric due to the presence of the double bond ring system and substitution in one of the benzene rings [9].

## 2. Therapeutic Usage of Thioxanthenes

Much like the other neuroleptics, thioxanthenes are prescribed for patients suffering from schizophrenia. In addition to this, some of the drugs in this group possess specific characteristics that justify their application in clinical medicine.

Table 2 includes a comparative summary of these three classes of neuroleptics, listing some of the common risks of antipsychotic medications [10,11,12,13,14,15,16,17,18,19].

The first thioxanthene, chlorprothixene, was found to have an excellent effect in schizophrenic patients [20]. Chlorprothixene had been used in the treatment of neuroses, not only due to its sedative and calming effects but also due to its low level of toxicity and side effects. Treatment with rather low doses results in favorable results in neuroses with anxiety, tension, insomnia, psychosomatic disorders, and depression. Chlorprothixene had also been found useful for treating alcoholics and alcohol psychoses [21].

Flupenthixol is the highly potent thioxanthene analogue of the phenothiazine fluphenazine [22]. Flupenthixol possesses a distinct anxiolytic property. At low and, sometimes, rather higher doses, this drug is effective against hallucinations and delusions. Flupenthixol manifests stimulating or activating properties in low doses; sometimes even apathetic patients show greater alertness. Schizophrenic patients treated with chlorpromazine, levomepromazine, or other thioxanthenes are sometimes given flupenthixol in low doses as aftercare. It can control psychotic symptoms without affecting the alertness or working ability of the patients [9,23]. The action of flupenthixol is much similar to that of trifluoperazine, but the former induces much less extra-pyramidal effects [24]. Flupenthixol is often prescribed for mood stabilization when psychiatric patients suffer from depressive neurosis. During the early 1970′s, intensive clinical studies repeatedly proved the excellent efficacy of flupenthixol over other neuroleptics in depressive patients [25,26]. Administration of this drug in low dosages does not usually produce side effects, however, sleep disturbances may occur in some patients who are treated after 5 pm [9].

Clopenthixol has a narrower field of application compared to chlorprothixene, although often it produces quick action on patients suffering from delusions, aggressiveness, destructiveness, impulsiveness, and even hallucination, and has also been proven to be definitely better than chlorpromazine for treating paranoid schizophrenics [27]. It produces highly satisfactory results in paranoids and catatonics [28]. Treatment with clopenthixol may start with a low dose followed by a gradual increase. However, therapy has to be continued on a regular basis with one tablet in the evening.

## 3. Pharmacological Properties of Thioxanthenes

Much like the phenothiazines, thioxanthenes exhibit varied pharmacological actions, peripheral as well as central. However, therapeutic uses of these compounds depend on their psychopharmacological activity. The neuroleptic potency of a synthesized thioxanthene depends on the structure of the side chain in position 9. The compounds with β-hydroxyethylpiperazinopropyl or β-hydroxyethylpiperidinopropyl side chains are more potent neuroleptically than those with a dimethylaminopropyl side chain [29]. The antagonistic effect against methylphenidate-induced stereotypes in mice was employed to determine the exact duration of neuroleptic action of a compound. It was found that the peak effect was between 2 and 6 h, and by 24 h, effects of all the compounds were gone [9,30].

Neuroleptics are known to block the dopamine-induced formation of cyclic adenosine monophosphate (cAMP). In an elaborate study, Iversen et al. (1974) [31] observed that *Z*-flupenthixol was the most potent neuroleptic among all the test phenothiazines, thioxanthenes, and tyrophenones. Interestingly, they noted that *E*-flupenthixol was completely inactive.

It is known that neuroleptics have to be administered to psychiatric patients on a long term basis. In 1974, Moller et al. [32] observed that the antagonistic effect of neuroleptics undergoes tolerance development in animal models after prolonged therapy. They reported that when rats were pre-treated with flupenthixol for 12 days, followed by 3 days withdrawal, the antagonistic potency against apomorphine stereotypes was decreased. This reduction in potency subsequently gradually disappeared.

Neuroleptics are known for their α-adrenolytic function. Between phenothiazines and thioxanthenes, the adrenolytic activity was found to be more prominent in the *Z*-isomers of chlorprothixene, flupenthixol, and clopenthixol, while chlorpromazine and fluphenazine (which has no *E/Z* center) revealed much less activity [33].

Nasrallah & Tandon (2013) [9] observed that there was a moderate reduction in adrenaline pressor response after administration of either chlorprothixene or flupenthixol to anaesthetized cats. Furthermore, in vagotomized cats, the carotid occlusion reflex was reduced. After administration of chlorprothixene in these animals, respiration remained unaffected, although initially there was a transient rise in respiratory minute volume, due to increased tidal volume. Intravenous infusion of chlorprothixene or flupenthixol in conscious dogs resulted in the fall of blood pressure, without any change in pulse pressure. Treatment of dogs exhibiting tachycardia with flupenthixol resulted in a normalization of heart rate for 15 min [9,14].

## 4. Bacterial Inhibitory Action of Thioxanthenes

Human bodies harbor a plethora of diverse communities of commensal, symbiotic, and pathogenic microorganisms, along with their genetic compositions, collectively known as the ‘microbiome’ [34]. The gastro-intestinal tract is the main location of the human microbiota. Recent ground-breaking studies in modern science are focusing on the role gut microbiota play in the pathogenesis of several medical conditions, particularly those related to the central nervous system. This new concept has been termed the ‘microbiota-gut-brain axis’ [35]. Bidirectional communication lines, effected by the neural, endocrine. and immune systems, tightly link the huge array of bacterial populations in the gastro-intestinal tract with the brain. Mediators of this axis include short chain fatty acids (e.g., butyrate), neurotransmitters (e.g., serotonin and γ-aminobutyric acid (GABA)), hormones (e.g., cortisol), and immune system modulators (e.g., quinolinic acid) [36]. Recently, the gut microbiome has been termed aptly as the ’psychobiome´ [37]. Several reports indicate the connection of gut bacteria with the development of neurodegenerative diseases, (like Alzheimer’s, Parkinson’s and Schizophrenia), with their associated cognitive decline [38,39,40,41]. Epidemiological researchers have also noticed an increase in depression in people taking antibiotics. Numerous *in vitro* and *in vivo* studies showcase the varying effects of widely used psychotropics on microorganisms. An expanding body of experimental evidence supports the notion that microbes can metabolize drugs and vice versa, that drugs can alter the microbial composition. In 1954, Geiger & Finkelstein [42] observed that neurological patients receiving chlorpromazine could be cured of tuberculosis much faster. Similar observations started to be reported from various other scientists: there are medicinal compounds used for therapy of non-infectious pathology which simultaneously possess antimicrobial action. All such compounds are collectively known as ‘non-antibiotics’ [43]. Although many of the non-antibiotics are tricyclic antidepressants [44], there are compounds that possess two benzene rings which are joined to each other by different structural moieties [45,46,47,48,49,50,51]. Structurally similar thioxanthenes were also reported to possess broad-spectrum antibacterial function. In 1987, Mortensen and Kristiansen [52] reported that different forms of clopenthixol possess moderate to powerful action against both Gram positive and Gram negative bacteria (Table 3). Of the two stereo-isomeric compounds, *E*-clopenthixol showed much greater antibacterial action than *Z*-clopenthixol. The MIC of *E*-clopenthixol, among most of the Gram-positive bacteria, was between 6.2 to 25 µg/mL. Despite that, in general, the Gram-negative bacteria were less sensitive: many could be inhibited at the 3.1 to 6.2 µg/mL level. Both *Corynebacterium* and *Listeria* were highly sensitive to *E*-clopenthixol. *Z*-clopenthixol was also active against Gram positive organisms and other test bacteria, but its MIC was definitely higher than those of *E*-clopenthixol.

Two main metabolites that are formed in humans after administration of clopenthixol are *N*-dealkyl-clopenthixol and clopenthixol sulfoxide. The former is a stereo-isomeric mixture of *N*-dealkyl-*Z*-clopenthixol and *N*-dealkyl-*E*-clopenthixol. This particular compound possessed a more powerful antibacterial function than administered *E*-clopenthixol, with an MIC often as low as 1.6 to 3.1 µg/mL. The other compound, clopenthixol sulfoxide, which is a mixture of *Z*- and *E*-clopenthixol sulfoxide, was practically inactive against most of the test organisms.

In an intensive study, Jeyaseeli et al. (2006) [53] reported highly potent antimicrobial action of another thioxanthene compound, flupenthixol. The results (Table 4) showed that the Gram-positive bacteria were highly sensitive to this thioxanthene as well, the MIC varying between the 5 and 50 µg/mL level. In a total of ten different genera of *Enterobacteriaceae,* many strains could be inhibited at 25 µg/mL while a few strains were resistant to flupenthixol. Strains of *Shigella* were mostly sensitive, while salmonellae were less sensitive. It may be pointed out here that the MIC of flupenthixol against *S. enterica* serovar Typhimurium NCTC 74 was 50 µg/mL. This particular strain was taken for animal experiments. All the test strains belonging to the genus *Klebsiella* were totally resistant to the drug, and most *Pseudomonas aeruginosa* strains were equally resistant. A large number of strains of *Vibrio cholerae* and *V. parahaemolyticus* were unable to grow in the low concentrations of the test drug.

Potent antimicrobial action of the phenothiazine derivative thioridazine, in combination with the antibiotic dicloxacillin, was reported by Poulsen et al. *in vitro* against Methicillin Resistant *Staphylococcus aureus* (MRSA) in 2013, 2018, and *in vivo* in 2014 [54,55,56]. Thioridazine showed antimicrobial action by itself, but in combination with dicloxacillin, the potency of thioridazine was increased, and the dose needed of both drugs was reduced remarkably [54,55]. This combinatorial beneficial effect seemed to be independent whether the thioridazine used was racemic or either of the two stereo isomers. Thioridazine administered alone, both *in vitro* and *in vivo*, did not affect bacterial growth at the dose chosen. Dicloxacillin administered alone kept growth in a steady state, but when the two drugs were combined, they inhibited growth and even killed the otherwise resistant bacteria [55]. The potential beneficial combinatorial treatment was shown *in vivo* using the nematode *Caenorhabditis elegans* (*C. elegans*) as a host model [56].

## 5. Bacteriostatic Action of Thioxanthenes

Jeyaseeli et al. (2006) [53] had conducted detailed antibacterial studies of flupenthixol. The drug was added to logarithmically growing *S. aureus* NCTC 6571, the amount being twice the value of its MIC. The number of viable bacteria fell from 10^8^ to 10^2^ in 6 h, following which there was no further reduction (Figure 1) in the number of viable cells up to 18 h. Parallel studies using *V. cholerae* 1347 revealed a very similar pattern of activity (Figure 2). This proved that flupenthixol exerts a bacteriostatic effect on both Gram-positive and Gram-negative organisms.

## 6. In Vivo Observation with Flupenthixol

In an elaborate study, Jeyaseeli et al. (2006) [53] used Swiss-strain white mice. Sixty animals were injected intraperitoneally with 0.1 mL of sterile saline, followed three hours later by a challenge of a 50 median lethal dose (MLD) of *Salmonella enterica* serovar Typhimurium NCTC 74 (Table 5). Another group of 30 mice each received 15 µg flupenthixol. Half of them, 20, were injected with the same amount of *S. enterica* intraperitoneally. Similarly, a further 30 mice were given 30 µg of flupenthixol in an identical manner, and again 20 of them were challenged with viable bacteria. In the control group that received saline, 48 out of 60 animals died within 100 h of the lethal challenge. However, a statistically significant protection was observed in the other groups of animals that received both the drug and the challenge, and this was confirmed by determining the bacterial load in various tissues (Table 6).

Jeyaseeli et al. (2012) [57] explored whether flupenthixol could efficiently augment the action of an antibiotic when tested in combination. Twelve bacterial strains belonging to various genera were selected for this study. These were all sensitive to the antibiotics: penicillin, ampicillin, chloramphenicol, tetracycline, streptomycin, gentamicin, erythromycin, and ciprofloxacin. Flupenthixol exhibited synergism with four out of the eight test antibiotics. Combining flupenthixol with penicillin and a disc diffusion assay system illustrated pronounced statistically significant synergism (*p* < 0.01). This was further confirmed with the help of the checkerboard method, and the fractional inhibitory concentration (FIC) index was found to be 0.375. Flupenthixol with penicillin were then tested in *in vivo* experiments in mice challenged with *Salmonella enterica* serovar Typhimurium NCTC 74. Statistical analysis of the mouse protection test suggested that this combination was highly synergistic (*p* < 0.001 by chi-squared analysis).

Similar augmentation of antimicrobial action was observed when flupenthixol was combined with other antibiotics, namely streptomycin, gentamicin and ciprofloxacin. The results of this study may thus provide alternatives for the treatment of problematic infections associated with *Salmonella* spp.

## 7. Effect of Thioxanthenes on Slow Growing Mycobacteria

Chlorpromazine came to be known from the intensive studies carried out by Laborit and his co-workers (1952) [58] on the usefulness of phenothiazines for artificial hibernation of surgical patients. The independent clinical trials by Sigwald and Bouttier (1953) [59] and Delay and Deniker (1952) [60] established chlorpromazine to be the best choice for psychiatric patients. However, in 1954, Geiger and Finkelstein [42] observed that psychiatric patients receiving chlorpromazine had a much faster recovery from tuberculosis. These observations paved the way to explore anti-tubercular properties in the other phenothiazines that had become available as antipsychotics or as anti-histamines. The detailed investigations by Popper and Lorian (1959) [61], Bourdon (1961) [62], Crowle et al. (1992) [63], Kristiansen and Vergmann (1986) [64], Molnar et al. (1977) [65], Ordway et al. (2003) [66], and Amaral et al. (1996) [67] repeatedly proved the efficacy of chlorpromazine against *M. tuberculosis.* In the past few decades, although a large number of studies were carried out on the anti-tubercular potentiality of chlorpromazine, several other phenothiazines were also found to be proficient in inhibiting the growth of *M. tuberculosis*
*in vitro* and also *in vivo*. Such drugs included promethazine and levomepromazine [65], trifluoperazine [68], methdilazine [44], and thioridazine [66,67,69,70,71].

In 1986, Kristiansen and Vergmann [64] were trying to determine anti-tubercular action in several thioxanthenes along with chlorpromazine and levomepromazine. Most of the test organisms were slow growing mycobacteria isolated at the Tuberculosis Department of the Danish National Serum Institute, Copenhagen, from patients with tuberculosis-like pathology. Additionally, they included *M. tuberculosis* strain No. 5 that was isolated from a patient prior to chemotherapy. The level of resistance to the drugs was determined by the agar dilution technique in oleic acid albumin agar, instead of Lowenstein-Jensen medium since the test drugs were sensitive to heat. Readings were noted after incubation at 35 °C for three weeks. The highest drug concentration (µg/mL), which permitted growth similar to that in the control, without drug, was recorded for every agent (Table 7). In this comparative study, clopenthixol (composed of *E*- plus *Z*-clopenthixol) exhibited potent anti-tubercular activity (Kristiansen and Vergmann, 1986) [64]. The authors further observed that chlorpromazine and levomepromazine possessed much less anti-tubercular activity compared to clopenthixol. Six other thioxanthenes tested were *Z*- or *E*-stereo-isomeric analogues of flupenthixol, clopenthixol, and chlorprothixene. Results revealed that *E*-flupenthixol was almost twice as potent as *E*-clopenthixol (Table 8). The rather small differences in the susceptibility of the test bacteria did not reveal any pattern of susceptibility to the *Z* or *E* forms of the same thioxanthene.

These results confirmed an earlier observation by Rajsner et al. (1975) [72] who reported on the anti-mycobacterial function of the thioxanthenes 6- and 7-fluoro derivatives of chlorprothixene.

Thus, this study showed that anti-tubercular action was demonstrated by both stereo-isomeric forms of the test compounds almost at identical levels. This is in contrast to the findings with respect to their action on Gram positive and Gram negative organisms where *E*-clopenthixol was more potent than *Z*-form [73]. Moreover, Kristiansen and Vergmann (1986) [64] found that the more resistant mycobacteria *M avium* and *M intracellulare* were rather sensitive to stereo isomers of thioxanthenes, but the only *M tuberculosis* strain (‘No. 5′) taken in their test was more sensitive to the test agents than *M avium* and *M intracellulare*. Although both stereo-isomers of thioxanthenes were almost equally active on mycobacteria, the *Z*-analogues may be excluded from further drug development because psychopharmacological studies have shown that they exert a neuroleptic effect [74], while *E*-analogues do not. With the help of further structural modifications, these drugs could be developed further for novel anti-tubercular drugs.

## 8. Effect of Thioxanthenes on Viruses and Eukaryotic Cells

In 1991, Kristiansen and her colleagues [75] observed that stereo-isomeric derivatives of thioxanthenes (the neuroleptic compound *Z*-chlorprothixene and the non-neuroleptic compounds *E*-chlorprothixene and *E*-flupenthixol) exhibit antiviral effects on Herpes simplex virus (HSV) and toxic effects on eukaryotic cells. The viral inhibition assay showed that *E*-chlorprothixene and *E*-flupenthixol both inhibited intracellular HSV expression in a dose dependent manner.

To determine cell growth inhibition, 24-well multidishes were seeded with 50,000 fibroblasts per well in 1 mL of growth medium, with or without test compounds [75].The tested compounds produced a concentration-dependent influence on the growth of the fibroblasts in cell cultures. Growth stimulation of 25–50% was noted at lower concentrations, while varying degrees of inhibition could be observed at higher concentrations. At 12.5 mg/L these compounds caused cell death. The study also included a toxicity test of the compounds. This was done by adding serially increasing amounts of them to flasks with a monolayer of fibroblasts. Attempts were then made to observe the cytotoxic effect daily and rate it semi-quantitatively under a light microscope (Table 9). *E*-chlorprothixene revealed no cytotoxic effect up to 4.4 µmol/L concentration while at 8.9 µmol/L there was 25–50% cytotoxic effect; however, at higher concentrations, 75–100% cells showed the toxic effect.

The inhibition of HSV expression by *E*-chlorprothixene and *E*-flupenthixol indicated that this effect was not linked to the neuroleptic effect, as the latter was observed only in the *Z*-derivatives of these compounds.

Thus, this study showed that at higher concentrations the cytotoxic effect could outmatch the antiviral action, while at lower concentrations both the activities could be observed. The critical concentration was 3 mg/L or 6–8.5 µmol/L, the amounts being much higher than plasma concentration tolerated by humans. In this way, combined antiviral and cytostimulatory actions could be observed in the same compounds. In a viral adherence assay, pre-incubation with the compounds could not protect the cells against the HSV infection. It has been suggested by Kristiansen et al. (1991) [75] that permeability changes in the cell membrane of eukaryotic cells may have interfered with a change in the immune response due to infections. The interference could be due to various factors like inhibiting the viral entry into the cells, inhibiting viral multiplication inside the cell, or by stripping the virus from infected eukaryotic cell. Thus, this study clearly indicated that these drugs possibly share a biological activity fundamental to both eukaryotic and prokaryotic cells. Studies on such unique compounds, that modify eukaryotic cell growth and also have antiviral properties, opens up the possibility to search for structurally similar compounds with even better dual functions for a better therapy for different viral infections in man.

## 9. In Vitro Modulation of Human Neutrophil by E-Clopenthixol

Phenothiazines are known to depress several functions of neutrophils like chemotaxis [76], oxidative metabolism [77,78] and phagocytosis [79]. These functions are known to depend on Ca^++^ fluxes [80,81]. Rechnitzer et al. (1985) [82] carried out a detailed study to determine chemotaxis of human peripheral blood polymorphonuclear leukocytes by the thioxanthenes Z- and E-clopenthixol.

To determine the chemotaxis of these compounds, Rechnitzer et al. (1985) [82] separated polymorphonuclear neutrophils from blood samples of normal human beings. There was no reduction in the number of viable neutrophils after 2 h of incubation up to 105 µM of the clopenthixols (Table 10). However, incubation with other thioxanthenes at 105 µM resulted in 22% to 33% cell death (data not shown here). At the same concentrations, when the cells were incubated for 2 h in the salt solution, there was 100% cell death in presence of these thioxanthenes.

*Z*- and *E*-clopenthixol at concentrations of 13 µM to 53 µM significantly enhanced neutrophil chemotaxis (directed migration) from 56% to 119%; however, chemokinesis (random migration) was not stimulated by these compounds.

Neutrophil chemotaxis towards casein was almost totally inhibited by 105 µM *Z*- and *E*-clopenthixol. Thus both *Z*- and *E*-clopenthixol exhibited a biphasic effect on human neutrophil chemotaxis (data not shown here). These stereo-isomers increased chemotaxis by two-fold at the level of peak-response. Such an enhancement was observed in a very wide range of concentrations. Clopenthixol-induced enhancement could possibly be due to cell-receptor activation by binding of the drug to the receptor on the cell surface and activation of Ca^++^ mobilization. Another possibility could be that these compounds interact hydrophobically with Ca^++^ dependent proteins. Such an interaction could result in physical alteration of the membrane fluidity as shown in chlorpromazine [81] and by the observation of a Fibonacci-series correlation between water-soluble-phenothiazine and water-insoluble phospholipids adducts [83]. Moreover, this effect is probably dependent on the levels of cholesterol, calcium, and, to a less extent, drug concentration [84]. The inhibition of chemotaxis and polymorphonuclear neutrophils by these drugs could be due to their action on the stabilization of membrane. The increase of chemotaxis of human neutrophils by *E*-clopenthixol is of substantial importance. since it possesses both antimicrobial and anti-plasmodial activities but is devoid of antipsychotic [85] and anti-hypersecretory properties [86].

## 10. Inhibition of HIV Replication by Thioxanthenes

Several studies during the 1990s showed that microorganisms that have higher potential for infecting nervous tissues are usually very sensitive to neurotropic drugs [87]. Viruses like polio, influenza, measles, and herpes, with greater chances of infecting brain and nervous tissues, can be affected by stereo-isomeric forms of thioxanthenes [75]. In 2000, Kristiansen and Hansen [88] reported on the antiretroviral activity of stereo-isomeric analogues of thioxanthenes. Expression of HIV antigens, p17 and p24, was determined with the assay method as described by Sarin et al. (1988) [89]. Six different stereo-isomers of thioxanthenes were tested along with amitriptyline and nortriptyline as anti-HIV drugs.

Kristiansen and Hansen (2000) [88] found out that, among all the thioxanthenes tested, only *Z*-flupenthixol and *E*-flupenthixol revealed anti-HIV activity at 2 mg/L and 10 mg/L respectively (Table 11). However, no such activity was demonstrated by a mixture of the above two compounds. The action primarily resided in *Z*-flupenthixol. Addition of a fluorine atom at position 6 in *Z*-flupenthixol, forming piflutixol, results in a loss of anti-HIV activity. Similarly, synthesis of zuclopenthixol from *Z*-flupenthixol also resulted in loss of the activity. Substitution of the CF3-group in position 2 of the same compound with an isoelectric chlorine atom resulted in formation of the same zuclopenthixol, which had no activity. The tricyclic antidepressants amitriptyline and nortriptyline produced rather weak anti-HIV activity at distinctly high concentrations. Other known antidepressant compounds produced weak to moderate anti-HIV activity at their respective non-toxic concentrations. Most of these drugs were found to be toxic at higher concentrations.

This study was carried out to locate and find out agents that can be given simultaneously along with standard anti-HIV drugs to infected patients suffering from dementia. Therefore, this study clearly indicated the remarkable possibilities of combining at least two potent anti-HIV thioxanthenes with known therapeutic drugs to treat HIV patients who had developed AIDS related dementia. Similar potential effects has been shown of some of the phenothiazines on SARS-CoV-2.

## 11. Antiparasitic Action of Thioxanthenes

During the latter part of nineteenth century, infection by malarial parasites became quite prevalent in various parts of the world, and hence search for antimalarial drugs went on in every continent. This resulted in the discovery of quinine, an extract from the bark of cinchona plant, way back in the 17th century in Europe [90]. The pharmaceutical compound chloroquine was developed by Bayer Laboratories in 1947. This led to rapid eradication of malaria from different parts of the world. However, soon after, this wonder drug became useless, because the parasite became resistant to chloroquine [91,92]. It needs to be pointed out here that as early as 1891 Guttman and Ehrlich [93], while trying to determine the therapeutic usefulness of methylene blue, very successfully demonstrated antimalarial action of this dye *in vivo*. With this background in mind, Kristiansen and Jepsen (1985) [94] initiated a study to search for antimalarial drugs from the neuroleptic drugs phenothiazines and thioxanthenes.

Kristiansen and Jepsen (1985) [94] obtained chlorpromazine, *Z*- and *E*-clopenthixol, in pure drug powder form. The test parasite was the known chloroquine-sensitive strain *Plasmodium falciparum* FCDL_1_. The IC_50_ value was calculated on the basis of 50% inhibition of *P. falciparum* multiplication (Desjardins’ ^3^H-hypoxanthine uptake assay) [95] in presence of a drug. This revealed that among the two thioxanthenes, *Z*-clopenthixol was more powerful than *E*-clopenthixol (Table 12) throughout the study. When tested against other bacterial strains, *E*-clopenthixol was more potent than *Z*-clopenthixol [96].

## 12. Efflux Pump Inhibition by Thioxanthenes

Efflux pumps are known to induce multidrug resistance in pathogenic bacteria, thereby resulting in complications in chemotherapy of microbial infections. The pumps help bacteria to evade effects of certain antibiotics. The reasons for multidrug resistances in the virulent organism *Pseudomonas aeruginosa* are due to presence of several three-component efflux systems that enable the bacterium to eject the antibiotics without any difficulty. The most extensively studied efflux system is MexAB-OprM, which, along with MexXY-OprM, is involved in intrinsic resistance to several antibiotics in *P. aeruginosa* [97]. Similar efflux pumps are also present in Gram positive bacteria, and they confer resistances to macrolides, tetracyclines, and fluoroquinolones [98]. In an extensive study, Kuroda et al. (2001) [99] elaborately described that *S. aureus* has a genome size of 2–8 Mb and possesses about 253 open reading frames encoding putative transport pumps (including the Nor A protein which is capable of translocating hydrophilic fluoroquinolones). Therefore, inhibition of function of these pumps would restore the action of antibiotics. Only a few chemical compounds, for example reserpine and verapamil, have been identified as inhibitors of efflux pumps like the Nor A pump [100].

The dopamine receptor antagonists phenothiazines and thioxanthenes are known to possess potent antimicrobial action [71]. Although these agents require rather large amounts compared to antibiotics to produce antimicrobial action *in vitro*, their tissue levels are usually several fold higher, and inhibitory concentrations can be achieved at the site of infection [101].

While the *Z*- stereoisomer of thioxanthenes is a highly potent neuroleptic, both forms possess antimicrobial action. However, the antibacterial effect is often greater in the *E* form [82]. Combination of a thioxanthene and an antibiotic sometimes has a synergistic result, as observed by Kristiansen et al. (1988) [102] and Jeyaseeli et al. (2012) [57]. Ford et al. (1989) [103] provided enough evidence that thioxanthenes can inhibit the action of eukaryotic efflux pumps including p-glycoprotein, making them useful for the treatment of drug resistant tumors. Although the exact mechanism by which the thioxanthenes promote antimicrobial activity of antibiotics is not yet understood fully, it has been suggested that such a phenomenon could be due to the inhibition of efflux pumps [104].

According to Kuroda et al. (2001) [99], *S. aureus* possesses a large number of chromosomally encoded multidrug resistant (MDR) efflux pumps, many of which have still not been characterized. Kaatz et al. (2003) [105] observed that inhibition of these pumps could be achieved by certain thioxanthenes and phenothiazines, resulting in reversal of resistance to several antibiotics. The thioxanthenes that they elaborately studied were the two geometric stereo-isomeric forms of flupenthixol. They used several strains of *S. aureus* possessing unique efflux-related MDR phenotypes. Both compounds possessed some intrinsic antimicrobial activity. However, in combination with certain efflux pump substrates (antibiotics) in *S. aureus* strains, these compounds produced additive or synergistic effects. They further observed that, in a particular strain of *S. aureus* that could over express the Nor A MDR efflux pump and in two other strains possessing non-Nor A- mediated MDR phenotypes, the IC_50_ value for the ethidium bromide (EtBr) efflux pump was much lower, being 4 to 15% of their original respective MICs. *E*-flupenthixol, being more active than *Z*-flupenthixol as an antimicrobial agent, was able to reduce proton motive force (PMF) by reducing the transmembrane potential.

Kaatz et al. (2003) [105] believed that the inhibitory action of MDR efflux pumps in *S. aureus* was multifunctional, including distribution of membrane energetics and a likelihood of a direct interaction with the transporters themselves.

Dey et al. (1999) [101] provided sufficient data to prove that a single amino acid change in human p-glycoprotein could affect the inhibitory activity of both *Z*- and *E*-flupenthixol, suggesting that these stereo-isomers could interact directly with these pumps. Kaatz et al. (2003) [105] suggested that the mechanisms by which thioxanthenes inhibit the efflux system by PMF-dependent pumps is multifactorial. Since these compounds produced an effect on the MICs of antibiotics in S. aureus and also on the efflux system of different substrates, there may be an interaction with the pump itself combined with reduction in transmembrane potential. According to Kaatz et al. (2003) [105], these thioxanthenes, particularly *E*-flupenthixol, may turn out to be a logical solution to produce a rather non-toxic broad spectrum bacterial efflux pump inhibitor.

## 13. Conclusions

A panoramic view of the properties of thioxanthenes reveals that they are multifunctional in nature. Release of dopamine in the mesolimbic pathway is known to be linked to the expression of psychosis. Tricyclic thioxanthene drugs are known to accumulate in the brain and block D_2_ receptors in the dopamine pathway in such a manner that the effect of the released dopamine is reduced.

Thioxanthenes, however, are given to patients, not only as antipsychotic agents, but also as antidepressants and anxiolytics. There are also numerous strong indications that they can influence the microbiota-gut-brain axis. In this group of drugs, the *Z*- and *E*-clopenthixol, along with flupenthixol, showed moderate to powerful antibacterial action *in vitro* and *in vivo*. One of the thioxanthenes, flupenthixol, has shown synergistic activity with several antibiotics in reducing the dose of antibiotics needed to treat to patients suffering from specific bacterial infections. Some other neuroleptics like *Z*- and *E*-stereo-isomeric analogues of flupenthixol, clopenthixol, and chlorprothixene have proved to be significantly anti-mycobacterial in nature.

This unique class of compounds has further been shown to possess antiparasitic and antiviral, including anti-HIV and anti-SARS-CoV-2, activities as well. Moreover, some of the compounds were observed to have combined antiviral and cytostimulatory actions. Several observations regarding this combined action suggest that permeability changes in the cell membrane of eukaryotic cells might interfere with a change in the immune response, with respect to infections. This interference would be conveyed in various ways, for example, by inhibiting the virus from entering the cell, by inhibiting intracellular multiplication of the virus by creating differences in the nutrition of the eukaryotic cell systems, or by stripping the virus from the infected cells. Thus, there is a possibility that the thioxanthenes may be affecting a physical or biological activity which is common to both prokaryotic and eukaryotic cell systems. Additionally, a particular drug, *E*-clopenthixol, can participate in modulation of human neutrophils *in vitro*. Efflux related multidrug resistance has been experimentally proven to be a significant means by which pathogenic bacteria can evade the antibacterial activity of some selected agents. Several bacteria including *Staphylococcus aureus* have been shown to possess numerous chromosomally-linked efflux pump genes. Intensive studies have significantly revealed that thioxanthenes are able to inhibit the action of these pumps, resulting in the restoration of antimicrobial activity of the antibiotics. Two stereoisomers of flupenthixol, when combined with multidrug resistant efflux pump substrates, produced synergistic antibacterial properties against Gram positive bacteria.

Novel drugs are often developed with their organotrophic effects in mind. The possibility that they may have a powerful effect on microorganisms living in and on patients is often overlooked. In a similar way, antibiotics are developed focusing on killing microorganisms, but they may also affect the host directly. Thus, the generally accepted clinical interaction model is only a special case of reality. A general applicable clinical model has to take into account all of the interactions, and reactions, between the drug, the organism, and all microorganisms present [106]. As described above, thioxanthenes have ‘non-antibiotic’ effects on endogenous microorganisms and, in addition, possess synergistic effects with classical antibiotics. Instead of a complication, drugs like thioxanthenes, showing double-edged sword effects, should be seen as a golden opportunity.

We are standing on the edge of a cliff facing what some have described as a potential antibiotic resistance pandemic. Whether we are swept off or step away depends on whether we harness new ways to deliver existing compounds, harnessing, for example, lipid nanocapsules [107], whether we use synthetic (stereo)chemistry to sharpen their effects by using aminated thioxanthenes [108], select new compounds from libraries created using novel synthetic pathways [108,109], or whether we can mathematically predict non-antibiotic candidates [110].

Novel, powerful systems which mimic the *in vivo* interaction between a microorganism and the host (represented by immortal cells, stem cells or even primary cells in mono or co-cultures) are being developed [111,112,113,114,115]. These systems will allow the rapid and efficient testing of the full effects of non-antibiotics in close to *in vivo* environments.

Thus, this unique class of compounds, the thioxanthenes, have been repeatedly found to possess a wide variety of activities. They are not only antibacterial, antiviral and antiparasitic in nature, but they are also able to modulate human neutrophils and take part in the inhibition of multidrug-resistant bacterial efflux pumps. Therefore, the antimicrobial properties of thioxanthenes can provide inexpensive therapy that may be of immense value for the less fortunate communities in the world. In this way, an entirely new avenue for the treatment of bacterial infections may open up, if and when the pharmaceutical industries are able to recognize the multifarious activities of thioxanthenes. This double-edged sword could result in the syntheses and practical application of novel formulations for human use. Since new antibiotics have not been discovered for decades, and since the spread of bacterial drug resistance has reached an uncontrollable stage, this is an opportunity which should not be ignored.

Research has demonstrated that some of the common side effects of some antipsychotic drugs, such as prolonged QT interval, are observed with only one of the two enantiomers [116,117]. This underlines the importance of having good procedures for the synthesis of the optically pure enantiomers, something that has been found to be unsatisfactory for the drug thioridazine. Antonsen S. has used an auxiliary-based strategy for the total synthesis of both enantiomers in high optical purity [118]. Ongoing studies have shown that the antimicrobial effects of these two enantiomers are similar [118], but further investigations are needed to determine whether the side effects are caused by one or both enantiomers. In situations where the side effects are caused by only one enantiomer, pure-enantiomeric non-antibiotic drugs will open up the possibility for better treatments of resistant bacterial infections.

## Figures and Tables

**Figure 1 molecules-27-00196-f001:**
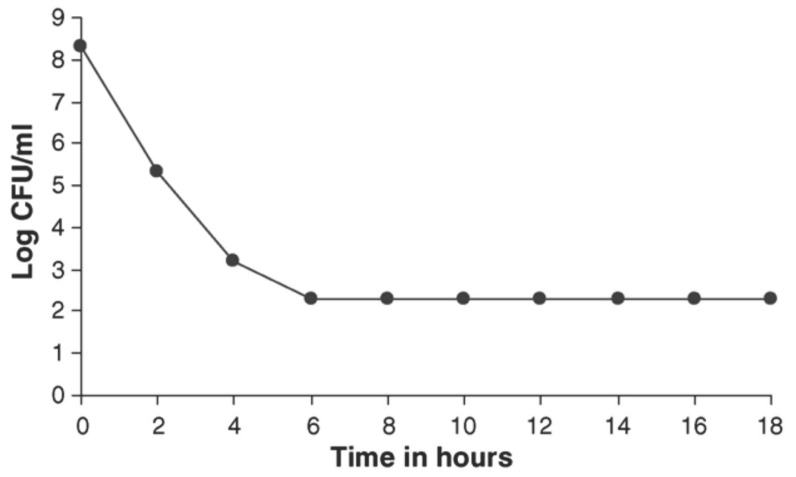
Bacteriostatic action of 20 µg/mL flupenthixol on *Staphylococcus aureus* NCTC 6571 (minimum inhibitory concentration of 10 µg/mL). Adapted from: Jeyaseeli et al. 2006 [53].

**Figure 2 molecules-27-00196-f002:**
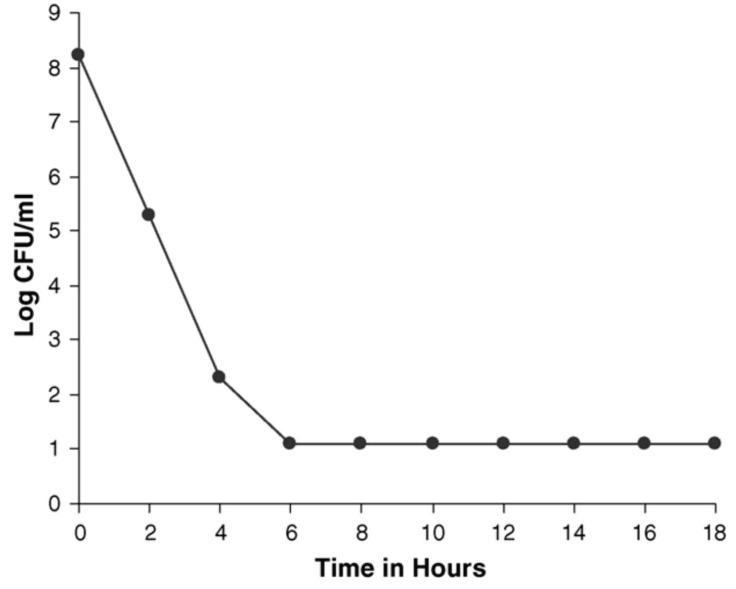
Bacteriostatic action of 20 µg/mL flupenthixol on *Vibrio cholerae* 1347 (minimum inhibitory concentration of 10 µg/mL. Adapted from: Jeyaseeli et al. 2006 [53].

**Table 1 molecules-27-00196-t001:** Chemical classification of antipsychotic drugs.

Class of Antipsychotics	Drugs and Their Chemical Structures
**1. Phenothiazines** 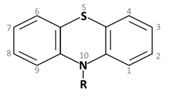 Phenothiazine basic ring structure	
a. Amino alkyl compounds: (Low/medium potency agents that can antagonize α_1_-adenoreceptors, histamine H_1_ receptors and muscarinic cholinergic receptors)	Chlorpromazine: 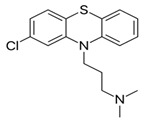
b. Piperidine compounds: (Low/medium potency agents and also muscarinic antagonist)	Thioridazine: 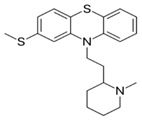
c. Piperazine compounds: (Medium/high potency agents)	Trifluoperazine: 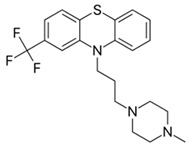
**2. Butyrophenones** 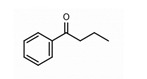 Butyrophenone basic ring structure (High potency agents)	Haloperidol: 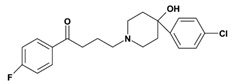
	Droperidol: 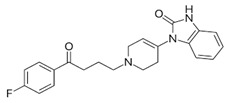
**3. Thioxanthenes** 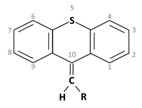 Thioxanthene basic ring structure (Medium potency agents)	Chlorprothixene: 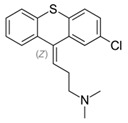
	Flupenthixol: 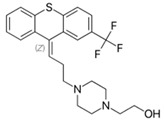
	Clopenthixol: 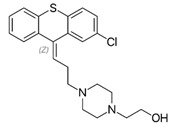

(Note: The terms “low/medium/high potency” indicates their potency in binding to the dopamine D_2_ receptor).

**Table 2 molecules-27-00196-t002:** Comparative summary of antipsychotic side effects.

Adverse Effects	Drugs						
	(1) Phenothiazines			(2) Butyro- Phenones	(3) Thioxanthenes		
	Chlorpromazine	Thioridazine	Trifluoperazine	Haloperidol	Chlorprothixene	Flupenthixol	Clopenthixol
**Extra Pyramidal Side Effects:** The muscle related side effects observed with antipsychotic medications are termed as ‘Extra -Pyramidal Side Effects’ or EPS [10]	Low	Low	High	Very high	In a comparative study it was observed that Parkinsonian symptoms were more often found with chlorpromazine than chlorprothixene [11]	Develops in high dosages, can be controlled by anti-parkinsonian drugs [12,13]	High
**Anti-cholinergic** **Effects:** This includes symptoms like urinary difficulties, constipation, dry mouth, blurred visions and may lead to cognitive impairments.	High	High	Low	Very low	Moderate	Low	Both clopenthixol and flupenthixol were found to have lower effect in comparison to chlorprothixene [14]
**Sedation:** This is common with antipsychotic medications and is dose dependent.	High	High	Low	Produces much lesser sleepiness and calming effect than chlorpromazine [15]	High	Low	Low
**Hypotension:** Antipsychotics commonly cause orthostatic hypotension, depending on the degree of α_1_ adrenoreceptor antagonism.	High	High	Low	Very low	High	Moderate	Treatment with clopenthixol is often associated with orthostatic hypotension [19]
**Other Effects:**	**Photosensitivity:** Chlorpromazine is known to induce photosensitivity and skin pigmentation [16]. An intensive study with phenothiazines and thioxanthenes on schizophrenic patients [17] reported that patients receiving chlorpromazine showed statistically significant changes in the lens and cornea while patients treated with thioxanthenes did not.				**Hyperprolactinemia:** Thioxanthenes cause high prolactin levels due to the blockade of prolactin inhibitory factors (PIF), that inhibits release of prolactin from the pituitary gland [18].		

**Table 3 molecules-27-00196-t003:** Antibacterial effect of two stereo-isomeric forms of clopenthixol and its metabolites.

Bacteria	No. Strains Tested	Range of MIC (µg/mL) Observed			
		*Z*-Clopenthixol	*E*-Clopenthixol	*N*-Dealkyl-Clopenthixol	Clopenthixol Sulfoxide
*Staphylococci/Micrococci*	4	12.5–25	12.5	3.1–6.2	>100
*Streptococci*	13	6.2–50	3.1–25	1.6–12.5	>100
*Corynebacteria*	3	12.5	3.1–12.5	1.6–3.1	>100
*Listeria/Erisipelothrix*	2	12.5–25	6.2–12.5	6.2	>100
*Clostridium/Propionibacterium*	2	50	12.5	3.1–12.5	>100
*Enterobacteriaceae*	19	25–>100	6.2–>100	6.2–>100	50–100
*Aeromonas/Pseudomonas*	7	50–>100	25–>100	12.5–>100	>100
Other Gram-negative bacteria	11	12.5–>100	1.6–>100	3.1–>100	100–>100

Adapted from: Mortensen and Kristiansen 1987 [52].

**Table 4 molecules-27-00196-t004:** In vitro antibacterial activity of flupenthixol.

Bacteria	No. of Strains Tested	No. of Strains Inhibited by Flupenthixol (µg/mL)						
		5	10	25	50	100	200	>200
*Bacillus* spp	6		1	5				
*Staphylococcus aureus*	84	9	30	31	12	2		
*Streptococcus* spp	4		1	2	1			
*Escherichia coli*	47		1	5	4	5	3	29
*Salmonella* spp.	15		1		5		1	8
*Arizona* spp.	1			1				
*Providencia* spp.	1		1					
*Proteus* spp	4						1	3
*Shigella* spp	26	2	11	1	1			11
*Pseudomonas* spp.	12			1	2			9
*Pasteurella septica*	1						1	
*Bordetella bronchiseptica*	1			1				
*Hafnia* spp.	1			1				
*Klebsiella* spp.	5							5
*Vibrio cholerae*	111	5	9	23	26	5	4	39
*Vibrio parahaemolyticus*	33	1	1	9		1	5	16
**Total**	**352**	**17**	**56**	**80**	**51**	**13**	**15**	**120**

Adapted from: Jeyaseeli L et al. 2006 [53].

**Table 5 molecules-27-00196-t005:** Determination of the *in vivo* protective capacity of flupenthixol in mice receiving a challenge dose of *Salmonella typhimurium* NCTC 74 in 0.5 mL nutrient broth.

Group	Drug Injected Per Mouse	Mice Died
Control (*N* = 60)	0.1 mL sterile saline	48
Group I (*N* = 20)	15 µg flupenthixol	3 *
Group II (*N* = 20)	30 µg flupenthixol	10 **

Note: None of the animals died when 15 µg of the drug alone was injected and one animal died when 30 µg of the drug was injected to two separate groups of mice (10 mice in each). * *p* < 0.001 according to *χ*^2^ test. ** *p* < 0.05 according to *χ*^2^ test. Adapted from: Jeyaseeli L et al. 2006 [53].

**Table 6 molecules-27-00196-t006:** Reduction in colony-forming units (CFUs) of *Salmonella typhimurium* NCTC 74 at 18 h following treatment with flupenthixol in heart blood and organ homogenates of mice.

Group	No. Mice Tested	Drug (µg/Mouse)	CFU/mL Count ^a^		
			Heart Blood	Liver	Spleen
I	5	Flupenthixol 15 µg	1.2–44 × 10^3^	6.5–73 × 10^3^	3.2–75 × 10^3^
Control	5	Saline	5.3–74 × 10^8^	8.5–50 × 10^8^	1.8–80 × 10^8^

^a^ Viable counts between two groups significant; *p* < 0.01 in 18 h samples (Student’s *t*-test). Adapted from: Jeyaseeli L et al. 2006 [53]

**Table 7 molecules-27-00196-t007:** The susceptibility of 10 slow-growing mycobacteria to 6 different *Z*- and *E*-thioxanthene derivatives.

Strains	Highest Drug Concentration Permitting Growth Quantitatively Similar to the Vehicle-Treated Control					
	*E*-Flupenthixol	*Z*-Flupenthixol	*E*-Clopenthixol	*Z*-Clopenthixol	*E*-Chlorprothixene	*Z*-Chlorprothixene
*M. tuberculosis* St. 5	6.25	6.25	12.5	6.25	6.25	12.5
*M. bovis* I 264	6.25	6.25	6.25	6.25	6.25	6.25
*B.C.G* T 1443	6.25	6.25	12.5	6.25	6.25	6.25
*M. marinum* T 2401	6.25	6.25	6.25	6.25	6.25	6.25
*M. scrofulaceum* T14447	12.5	12.5	12.5	12.5	12.5	12.5
*M. szulgai* 908	6.25	6.25	12.5	12.5	12.5	12.5
*M. xenopi* E 1613	6.25	12.5	12.5	12.5	12.5	12.5
*M. avium* T 10350	25	25	25	25	25	25
*M. intracellulare* ATCC 23432	6.25	6.25	12.5	12.5	12.5	12.5
*M. intracellulare* E 48067	12.5	12.5	12.5	12.5	12.5	25

Adapted from: Kristiansen and Vergmann 1986 [64].

**Table 8 molecules-27-00196-t008:** The susceptibility of 10 slow-growing mycobacteria to 6 different *Z*- and *E*-thioxanthene derivatives.

Inhibitory Agents	Number of Strains with Growth Similar to Vehicle Treated Controls					
	Drug Concentration (µg/mL)					
	<6.25	6.25	12.5	25	50	100
*E*-flupenthixol	10	9	3	1	0	0
*Z*-flupenthixol	10	9	4	1	0	0
*E*-clopenthixol	10	10	8	1	0	0
*Z*-clopenthixol	10	10	6	1	0	0
*E*-chlorprothixen	10	9	6	1	0	0
*Z*-chlorprothixen	10	9	7	2	0	0

Adapted from: Kristiansen and Vergmann 1986 [64].

**Table 9 molecules-27-00196-t009:** Cytotoxic effect of thioxanthenes.

Concentration (mg/mL)	Chlorprothixene HCI (mol.wt. 352)			*E*-Flupentixol (mol.wt. 508)	
	µmol/L	CTE		µmol/L	CTE
		*Z*-	*E*-		
0.3	1.1	*	*	0.8	*
0.7	2.2	*	*	1.5	*
1.56	4.4	0	0	3.1	0
3.13	8.9	0	++	6.2	+
6.25	17.6	0	++++	12.3	++++
12.5	35.5	0	++++	24.6	++++
25.0	71.0	0	++++	49.2	++++

CTE: cytotoxic effect: * = not tested; 0-no effects; + = <25% toxic effect; ++ = 25–50% toxic effect; +++ = 50–75 % toxic effect; ++++ = 75–100% toxic effect. Adapted from; Kristiansen et al. 1991 [75].

**Table 10 molecules-27-00196-t010:** Effect of Z- and E-clopenthixol on human neutrophil viability, determined by trypan blue dye exclusion after ½, 1, and 2 h incubation at 37 °C in GBSS containing 0.5% human serum albumin. Results are mean percentage of viable cells from 2–3 experiments.

*Z*-Clopenthixol Concentration in µm (µg/mL)	Percentage of Viable Cells			*E*-Clopenthixol Concentration in µm (µg/mL)	Percentage of Viable Cells		
	½ h	1 h	2 h		½ h	1 h	2 h
105 (50)	86	77	67	105 (50)	89	88	78
53 (25)	94	91	90	53 (25)	99	99	98
26 (12.5)	98	97	99	26 (12.5)	96	97	97

Adapted from: Rechnitzer et al. 1985 [82]. *n* = 2–3.

**Table 11 molecules-27-00196-t011:** Effect of neuroleptic drugs on HIV-1 replication.

Compound ^a^	Concentration ^b^ (mg/L)	% Inhibition of HIV Expression ^c^			
		Syncytia	P^17^	P^24^	RT
*Z*-flupenthixol	0.08	0	0	0	0
	0.4	0	0	0	0
	2.0	34	49	43	57
*E*-flupenthixol	0.1	0	0	0	0
	1.0	0	0	0	0
	10.0	43	66	67	67

^a^ ‘Flupenthixol’; ‘zuclopenthixol’, ‘chlorprothixene’, ‘*Z*-piflutixol’, ‘*E*-piflutixol’, and ‘citalopram’ showed no activity at the concentrations tested. ^b^ Minimum concentration of selected neuroleptic drug that significantly affects a parameter of HIV infectivity. ^c^ Inhibition of syncytial formation, p^17^ p^24^ and RT (reverse transcriptase) expression were carried out as described in Kristiansen and Hansen 2000 [88].

**Table 12 molecules-27-00196-t012:** Percentage inhibition of ^3^H-hypoxanthine uptake in *Plasmodium falciparum* at various concentrations of *Z*-clopenthixol and *E*-clopenthixol.

Compound	Percent Inhibition of *Plasmodium falciparum* In Vitro							
	Drug Concentration (µg/mL)							
	0	0.039	0.078	0.156	0.312	0.625	1.25	2.5
*Z*-clopenthixol	0	0	0	17	30	76	99	100
*E*-clopenthixol	0	0	0	9	12	36	83	95

Adapted from: Kristiansen and Jepsen 1985 [94].

## Data Availability

The source of the presented data is the references which have been cited.

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
