# Peer review of "A Double-Edged Sword: Thioxanthenes Act on Both the Mind and the Microbiome"

_molecules, 2021, doi:10.3390/molecules27010196_

Round 1
Reviewer 1 Report
A double-edged sword: thioxanthenes act on both the mind and the microbiome
This is an interesting and detailed review of the pharmacological properties of thioxanthenes that reveals a great diversity of the therapeutic use of these substances. A detailed analysis of the chemical structure of thioxanthenes related to their therapeutic activity and their adverse effects compared to phenothiazines and butyphenones is presented. These drugs have "non-antibiotic" effects on endogenous myco-organisms and have synergistic effects with the classic antibiotics. This article is well presented with an extensive up-to-date bibliography. Its reading is easy and understandable.
Author Response
Thank you very much for your positive comments. In accordance to the comments from Reviewer 2, we have made changes. Please see the attachments.

Reviewer 2 Report
The comments regarding the manuscript entitled: A double-edged Sword: “Thioxanthenes Act on Both the Mind and the Microbiome” are presented below.
The subject of this article is interesting and could be useful for the researchers involved in this field of science. Due to this fact I consider that supplementary information and improvements are necessary.
Taking into consideration the observations below I recommend the publication of this manuscript after major revision.
In the article the comparation with phenothiazine derivatives is always presented, so why phenothiazine it is not mentioned in the title?
The literature is old, I found an article from 2019, four from 2017, three from 2016. The question is if the subject it is not interesting for the researcher, or the literature was not exhaustively investigated?
Line 69: Due to structural similarity, the antipsychotic activities of flupenthixol and clopenthixol are quite similar to the phenothiazine class (also known as the ‘piperazine group’).
Comment: From the structural point of view the Phenothiazine could not be considered in the “piperazine group” because the phenothiazine is a 1,4-thiazine derivative, not piperazine.
Line 130: However, it is possible to prolong the activity by substituting F in position 6.
Comment: Please be more specific, regarding the substituents able to prolong the activity.
Line 143: Neuroleptics are known for their α-adrenolytic function. Between phenothiazines and thioxanthenes, the adrenolytic activity was found to be more prominent in the cis-isomers of chlorprothixene, flupenthixol and clopenthixol while chlorpromazine and fluphenazine revealed much less activity.
Comment: The recommended IUPAC nomenclature is Z, E and not cis trans in the case of the presented compounds.
Comment: Please rephrase de sentence, in the structure of chlorpromazine and fluphenazine we do not have C=C double bond so they do not present Z/E isomers like cis-isomers of chlorprothixene, flupenthixol and clopenthixol.
Line 168 Several reports indicate the connection of gut bacteria with the development of neurodegenerative 169 diseases, (like Alzheimer’s, Parkinson’s and Schizophrenia), with their associated cognitive decline.
Comment: Please indicate the scientific literature
Line 180: Although many of the non-antibiotics are tricyclic phenothiazines[40] there are compounds that possess two benzene rings which are joined to each other by different structural moieties
Comment: The phenothiazine has two benzene rings connected by nitrogen and sulphur atoms and even though it is a tricyclic structure the ‘tricyclic phenothiazines’ sounds odd.
Line 185-202& all document: Of the two stereo-isomeric compounds, trans(E)-clopenthixol, showed much greater antibacterial action than cis(Z)-clopenthixol
Comments: Only E/Z nomenclature is suitable, not trans(E) or cis (Z)
Line224: the two stereo chemical variants.
Comment: stereo isomers not chemical variants
Figure 2: Mechanism of action of 20 μg/mL flupenthixol on Vibrio cholerae 1347 (minimum inhibi tory concentration of 10 μg/mL. Adapted from: Jeyaseeli et al 2006
Comment: The figure 2 does not present a mechanism of action, it presents only the decrease of cells viability in time
In the paragraph “Mechanism of antibacterial action of thioxanthenes” there is not presented information related with the Mechanism of antibacterial action…, the mechanism is missing.
Line 348: To determine cell growth inhibition, 24-well multidishes were seeded with 50,000 fibroblasts per well in 1 ml of growth medium, with or without test compounds. When 349 the cells in the control wells formed a monolayer, the multidishes were washed, fixed, 350 stained with crystal violet and visualized in a spectrophotometer at 570 nm[…
Comment:
In a review article I do not consider adequate to present a detailed experimental protocol; this is only an example but there are many others in the manuscript.
Author Response
Thank you for your constructive comments. We have made the suggested changes so please see the attachment for our Cover letter

Round 2
Reviewer 2 Report
The article can be published in its current form.
